# Spatial multi-omics of human skin reveals KRAS and inflammatory responses to spaceflight

Jiwoon Park [1,2], Eliah G. Overbey [1,2], S. Anand Narayanan[3], JangKeun Kim [1,2], Braden T. Tierney [1,2], Namita Damle[1], Deena Najjar [1], Krista A. Ryon[1], Jacqueline Proszynski[1], Ashley Kleinman[1], Jeremy Wain Hirschberg[1], Matthew MacKay[1], Evan E. Afshin[1], Richard Granstein[4], Justin Gurvitch [1], Briana M. Hudson[5], Aric Rininger[5], Sean Mullane[6], Sarah E. Church [5], Cem Meydan [1,2], George Church [7,8], Afshin Beheshti [9,10], Jaime Mateus[6] & Christopher E. Mason [1,2,11] ✉

Spaceflight can change metabolic, immunological, and biological homeostasis and cause skin rashes and irritation, yet the molecular basis remains unclear. To investigate the impact of short-duration spaceflight on the skin, we conducted skin biopsies on the Inspiration4 crew members before (L-44) and after (R + 1) flight. Leveraging multi-omics assays including GeoMx™ Digital Spatial Profiler, single-cell RNA/ATAC-seq, and metagenomics/metatranscriptomics, we assessed spatial gene expressions and associated microbial and immune changes across 95 skin regions in four compartments: outer epidermis, inner epidermis, outer dermis, and vasculature. Post-flight samples showed significant up-regulation of genes related to inflammation and KRAS signaling across all skin regions. These spaceflight-associated changes mapped to specific cellular responses, including altered interferon responses, DNA damage, epithelial barrier disruptions, T-cell migration, and hindered regeneration were located primarily in outer tissue compartments. We also linked epithelial disruption to microbial shifts in skin swab and immune cell activity to PBMC single-cell data from the same crew and timepoints. Our findings present the inaugural collection and examination of astronaut skin, offering insights for future space missions and response countermeasures.

During both short- and long-duration spaceflight, the human body is exposed to various factors that are unique to the space environment, including weightlessness (e.g., microgravity), increased radiation, isolation, and confinement[1,2]. Due to the unique remote and spaceflight environment, lifestyle and hygienic practices are different from an Earth setting and often simplified due to the minimal resources and space available in a spacecraft (e.g., lack of showers, restrooms, limited clothing available)[3]. Moreover, significant physiological adaptations and overall shifts in biological homeostasis occur from spaceflight exposure, including musculoskeletal deconditioning, cardiovascular fluid shifts, vision loss (e.g., spaceflight-associated neuro-ocular syndrome), metabolic shifts, and mitochondrial and immune dysfunction[4,5]. Several studies have identified the effects of spaceflight on the musculoskeletal and cardiovascular systems, including fluid redistribution, orthostatic intolerance, changes in heart rate, blood pressure, cardiac output, arterial stiffening, bone loss, atrophy of the striated and cardiac muscle groups, and increased risk of bone fracture[1,6–12].

The body's largest organ, the skin, is another key physiological system sensitive to environmental changes. Human skin serves multiple functions, including physical and immunological protection, supporting microbiota (containing bacteria, archaea, fungi, and viruses) homeostasis, thermoregulation, fluid retention, and metabolism[13–15]. Also, since the skin is the outermost layer of the body and exposed to the direct, external environment, it is usually the first substrate affected by environmental changes an individual experiences, including environmental adaptations resulting from spaceflight. The skin serves as a unique organ for investigating the impacts of spaceflight on the human body as it encompasses a range of physiological elements, from surface microbiota and connective tissue to vasculatures, follicles, and the nervous and immune systems, alongside various skin cell types. It offers a comprehensive platform to understand the multifaceted effects of spaceflight. Unfortunately, few studies exist on the spaceflight environment's impact on the skin, and observations to-date show that spaceflight may cause epidermal hypersensitivity and irritation, rarefaction of the cutaneous fiber network, and impaired epidermal proliferation and repair[16–20]. Nonetheless, the skin remains an understudied organ in the field of space biology; to date, fewer than ten astronaut case evaluations have been conducted[3]. Furthermore, only a few existing studies have examined transcript-level changes, and none of these studies has contextualized spatial or whole transcriptome gene expression or investigated the multiple omics components of skin adapting from spaceflight exposure[3,21,22].

Our study investigates skin samples collected from the SpaceX Inspiration4 mission and, for the first time, explores the spatial-transcriptomic characterization of short-duration spaceflight's impact on skin tissue. We comprehensively profiled skin microenvironment changes in response to spaceflight by performing a multi omics analysis using 4 mm skin punch biopsies from the crew members ($n = 4$) 44 days before launch (L-44) and 1 day after return (R + 1) of the 3-day mission. Tissue location specific gene expression levels were quantified using Nanostring GeoMx™ Digital Spatial Profiler (DSP) system across four regions of interest (ROI) targets, Outer Epidermis (OE) and Inner Epidermis (IE), Outer Dermis (OD), and Vasculature (VA) compartments. From immunofluorescence images, we identified epidermis, dermis, and vascular area and subdivided epidermis ROIs into outer and inner layers, which corresponded to granular and spinous layer (OE) and basal layer (IE). In total, we analyzed 95 astronaut skin ROIs using the human whole-transcriptome atlas (WTA, 18,676 genes). We also correlate these transcript level findings with matched metagenomic and metatranscriptomic data from skin swabs obtained prior to biopsy, as well as single-cell sequencing data (10X multiome single-cell ATAC and gene expression (GEX) characterizations) from isolated human peripheral blood mononuclear cells (PBMCs).

## Results

### Transcriptome-wide changes in response to spaceflight

To understand the impact of spaceflight to skin and tissue microenvironment, paired 4 mm skin punch biopsies from Inspiration4 crew members' upper arms were used for pathology evaluation and spatial transcriptomics profiling (Fig. 1a and Supplementary Fig. 1). In total, 95 ROIs were collected across 16 slides for processing, with the GeoMx whole transcriptome profiling probe set (18,422 probes). Based on imaging we selected four region types of interest, including the outer epidermis, inner epidermis, outer dermis, and the vasculature (OE, IE, OD, and VA). We also performed a skin histopathology analysis from the biopsied samples, which showed no significant abnormalities or changes in tissue morphologies or gross architecture (Supplementary Fig. 2).

From GeoMx spatial transcriptomics analysis, unsupervised clustering of all ROIs showed large clustering around compartmental identities. Slight shifts in response to spaceflight, and batch effects from both technical and biological replicates were not apparent after

normalization (Fig. 1b and Supplementary Fig. 3a). Differential gene expression analysis comparing post-spaceflight to pre-spaceflight samples found significant upregulation in 95 genes (log2FC > 0 and $q$ value < 0.05 by DESeq2) including *ARHGAP31, GALNT9, CPNE2, NMB, GPR50, CLDN2, OOSP2*, and downregulation in 121 genes (log2FC < 0 and $q$ value < 0.05 by DESeq2) such as *AP3B1, LMNA, COL6A2, VIM, HLA-B, PPP1CB, PABPC1* (Fig. 1c and Supplementary Data 1). Furthermore, proteins associated with cell junctions and extracellular matrices—particularly those from vimentin (*VIM*) and keratin (*KRT*) family—were the primary transcripts lost based on the DEG analyses.

Pathway analysis of these differentially expressed genes (DEGs) revealed statistically significant enrichment in kirsten rat sarcoma viral oncogene homolog (KRAS) signaling pathways, while transcripts associated with cell junctions and protein (i.e., apical junction, unfolded protein response) decreased (Fig. 1d and Supplementary Data 2). From expression levels, cell type composition for each ROI was estimated and compared across timepoints. We also observed statistically significant decreases in the cell type associated gene signatures of the major skin cell types and immune cells (e.g., melanocyte, pericyte, fibroblast, and T cells) (Fig. 1e).

### Region-specific expressions and spaceflight-induced changes

We then investigated region-specific expression changes across pre- and post-spaceflight samples for each ROI type label (OE, IE, OD, and VA). OE and IE regions were selected based on and corresponds to stratum granulosum and spinosum/basal, respectively. OD ROIs were selected by capturing a minimum of 200 cells inside of the basal cell layer (therefore mostly papillary layer), while VA ROIs were collected based on epithelial (FAP) and fibroblast (αSMA) staining (Fig. 1a). We observed transcripts specific to each ROI label and timepoint (Supplementary Fig. 3b, c).

For each ROI type, differential gene expression analyses were performed comparing postflight samples relative to preflight samples (Fig. 2a and Supplementary Data 1). For example, we found that the decrease in transcripts related to fibroblast and junction genes (e.g., *DES, ACTA2, TLN1, TAGLN*) specifically near the vasculature sites (VA). Loss of *KRT14* as well as other keratin family transcripts (*KRT1, 5, and 10*) were found predominantly in the dermal layer (OD). Taking the intersections of these DEGs to identify unique and overlapping genes across ROI types, we confirmed that most of the gene overlaps occur within ROI types that are relatively close to each other (i.e., VA and OD) (Fig. 2b). In particular, changes in *AP3B1*, a transcript related to granule formation, cytokine production, and inflammatory responses, were found in multiple comparisons (overall, OE, and OD) and was orthogonally validated with another technology, RNA scope (Supplementary Fig. 4a–c)[23]. In the inner layers of the tissue (OD and VA), we found overlapping DEGs related to stress and growth factor associated pathways, such as *COL6A2, CRKL, HLA-B*.

Gene set enrichment analysis (GSEA) revealed the consistent increase of KRAS signaling and inflammatory responses across all regions while specific immune pathways such as Interferon alpha and gamma response showed positive enrichment only in epidermal regions (OE and IE) (Fig. 2c and Supplementary Data 2). Pathways such as DNA repair, apoptosis, and UV response, reactive oxygen species were enriched only in the OE. We observed downregulation in genes involved with mitochondrial metabolism (e.g., myc target genes and oxidative phosphorylation) across all regions, particularly stronger in IE and OD ROIs. Also, the myogenesis pathway and EMT-related genes showed stronger decrease in enrichment scores in the VA ROIs, underscoring the region- and layer-specific responses to spaceflight. Comparing the pathway-level changes to blood sequencing datasets from the same mission and previous mission (NASA Twin Study, although with different duration of exposure), we found consistent changes in pathways such as KRAS signaling, epithelial-to-mesenchymal transition, and angiogenesis (Supplementary Fig. 4d)[5].

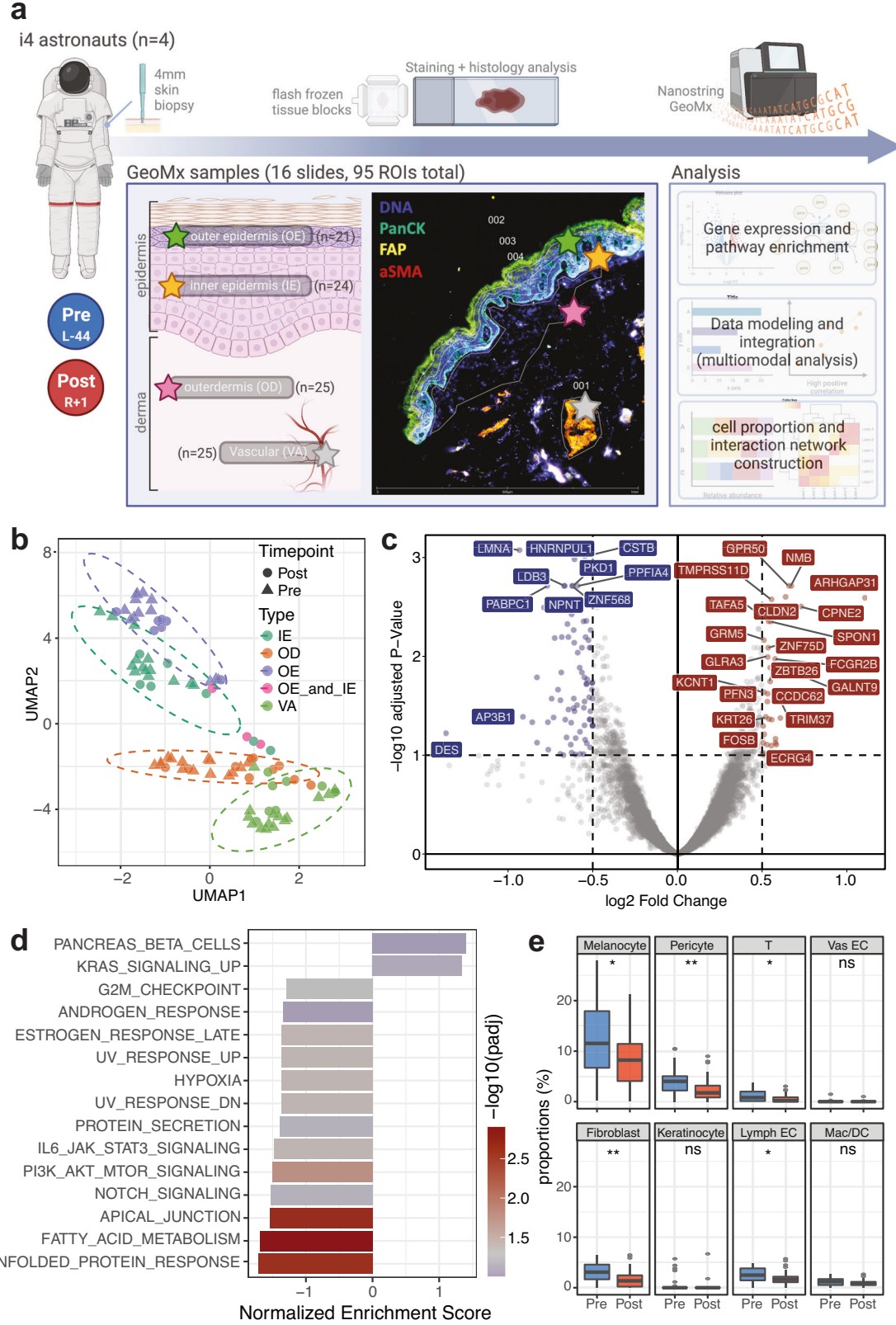

**Fig. 1 | Experimental design and overall pre- and post-timepoint comparisons.**
**a** Experimental design and workflow with representative tissue staining images (created with BioRender.com), **b** Uniform Manifold Approximation and Projection (UMAP) of all ROIs collected, **c** Volcano plot of overall post- vs. pre-spaceflight DEGs (using DESeq2 method), **d** Pathway enrichment analysis comparing DEGs from pre- and post-spaceflight skin tissues, visualizing normalized enrichment scores of MSigDB Hallmark pathways, and **e** Cell proportion comparisons between pre- and post-spaceflight samples (ns non-significant, *$p \leq 0.05$, **$p \leq 0.01$, ***$p \leq 0.001$, and ****$p \leq 0.0001$ by Wilcoxon test, two-sided; boxplot shows median/horizontal line inside the box, the interquartile range/box boundaries, whiskers extending to 1.5 times the interquartile range, and outliers as individual points outside the whiskers; exact $p$ values are included in the Source Data). Source data are provided as a Source Data file.

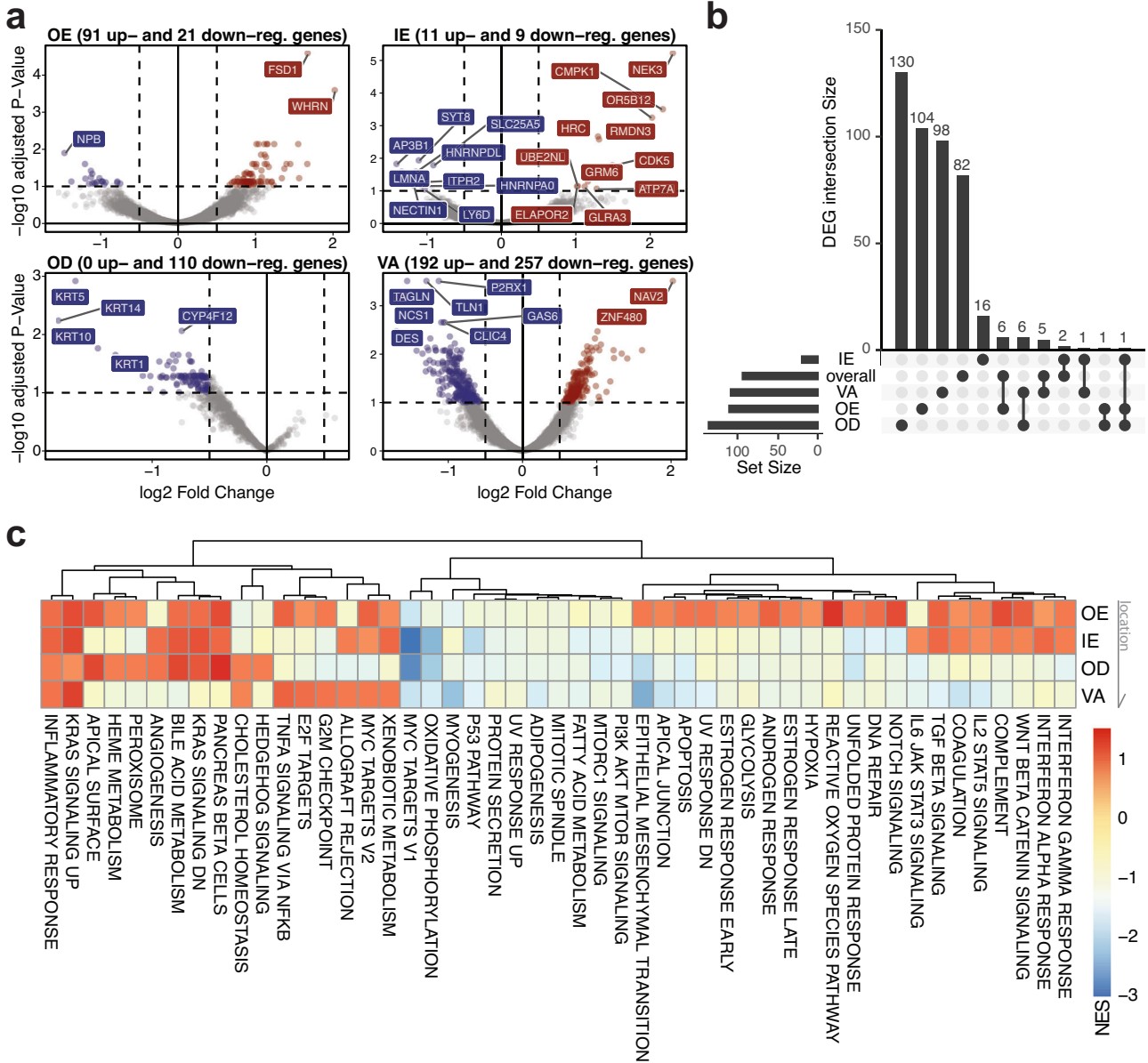

**Fig. 2 | Region-specific gene expression profile and cellular composition changes. a** Volcano plot showing DEGs by ROI types−OE, IE, OD, and VA respectively; the number of DEGs were determined by cutoffs of adjusted $p$ value < 0.1 and |log2FC| > 0.5 (using DESeq2 method), **b** UpSet plots comparing the intersections of region-specific DEGs, **c** Hallmark, non-germline gene set enrichment analysis across four ROI types; NES Normalized Enrichment Scores; Arrow indicates tissue locations, where OE is the outermost layer and VA is the innermost layer. Source data are provided as a Source Data file.

In addition to differential analyses, we also found that the marker genes reported to be specific to each skin layer and cell type corresponded to the expression levels in each ROI type and were consistent with the previous findings (Supplementary Fig. 3b, c)[24–26]. Based on the reference datasets, deconvolved cell type abundances were compared across ROI types and timepoints (Supplementary Fig. 5a). We found a loss of melanocyte related gene signatures specifically in the middle layers (IE and OD), not in the outermost region (OE) or vascular region deeper in the dermal layer (VA). On the contrary, fibroblast related gene expressions were decreased across all regions except for the outermost epidermal layer (OE). Although fibroblast is an unanticipated cell type in the epidermis ROIs, decreased fibroblast signature could indicate loss or damage of cellular and matrix interactions, consistent with previous reports highlighting the role of fibroblasts with epidermal regeneration (Supplementary Fig. 5b, c)[27,28].

**Epithelial barrier disruption and regeneration observed in post-spaceflight samples**

To investigate the phenotypic impact of spaceflight, we then focused on genes and pathways related to skin barrier formation, disruption, and regeneration. From the pathway analysis, we found enrichment changes in apical junction, UV stress response, hypoxia, and TGFβ signaling (Fig. 2c and Supplementary Data 2). Specifically, we observed a decrease in filaggrin (*FLG*) expression, a gene related to skin barrier function and plays a crucial role during epidermal differentiation by controlling interactions across cytoskeleton components, in postflight relative to preflight samples[29]. The decrease of *FLG* was the most evident in the OE region (Supplementary Data 1). Related to this observation, we also observed decreases in transcripts such as *HAS1, HAS2, HAS3, OCLN, CLDN, TGM2* in the OE region (Fig. 3a).

The decrease in protein production and response potentially are connected to decrease in keratinocyte and increase in immune

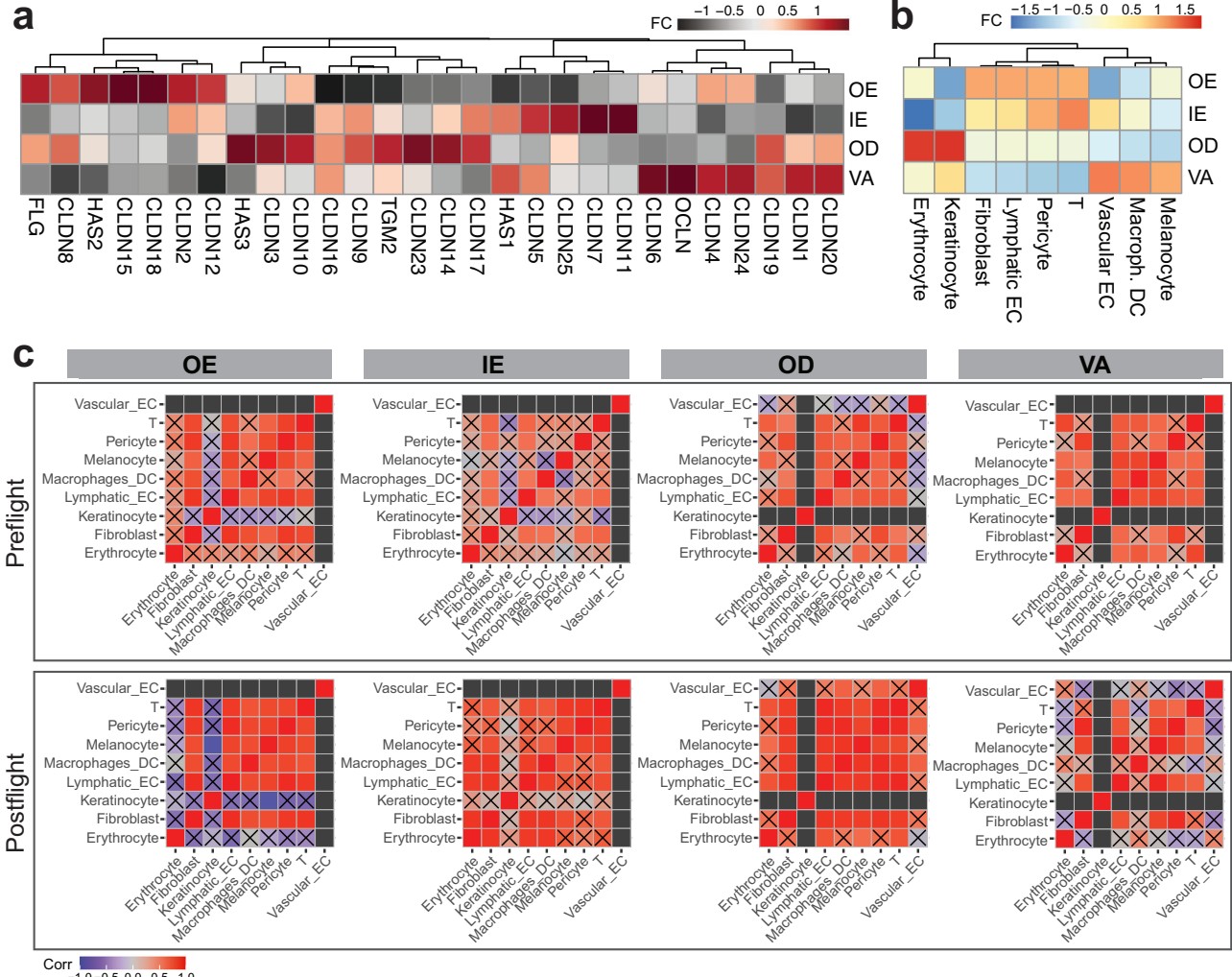

**Fig. 3 | Cellular microenvironment changes from spaceflight by tissue region.**
**a** Gene expression changes of interest, **b** fold change of proportions in post-flight samples relative to pre-flight samples, by compartments, **c** cell type correlation matrix changes. Black boxes represent undetermined spots (due to minimal cell counts); boxes with X marks represent correlations that did not pass statistical testing ($p$ value < 0.05, Pearson correlation, two-sided). Source data are provided as a Source Data file.

signatures (potentially related to interactions with T cells and fibroblasts) in OE region ROIs (Fig. 3b)[30]. Although weaker, the IE region shows a similar trend of cell proportion fold changes. Specifically, among fibroblast populations we also found that gene signatures of reticular fibroblast increased in postflight samples while there were no statistically significant changes in papillary fibroblast, suggesting disruptions in regeneration processes (Supplementary Fig. 5b, c)[31,32]. Taking co-occurrence of the proportion changes, cellular interactions within the ROIs were estimated. While cluster disruption was relatively minimal, an increase in melanocyte-macrophage interactions were found in the epidermis (OE and IE) ROIs (Fig. 3c). In addition, expression changes related to vascular and lymphatic endothelial cells and pericytes varied across the skin regions. The most pronounced cell signature changes were seen in the OE and VA compartments. In the OE compartment, we observed an increase in signatures related to lymphatic endothelial cells, potentially indicating the changes in the skin's vascular and immune system (Fig. 3b). While blood and lymphatic capillaries are not typically found in the epidermis, these adaptations may be suggestive of a wound-healing phenotype with the skin, which is supported by our results showing increased damage, inflammation, apoptosis, ROS, hypoxia, angiogenesis, TGF-beta expression, etc., in the epidermis (Fig. 2c)[33,34]. On the other hand, in the VA compartment,

there was an increase of gene signatures related to blood endothelium and decrease in lymphatic endothelium, also associated with vascular remodeling events.

## Skin-microbiota interaction changes during spaceflight

To test whether immune activation and epithelial barrier disruption can be explained with external environmental change, we performed metagenomics and metatranscriptomics analysis on the skin swabs collected right before biopsies (Supplementary Fig. 6a). After assignment of taxonomic labels to DNA sequences, we identified 826 bacterial and 9819 viral species with non-zero counts from metagenomics analysis, and 88 bacterial and 1456 viral species from metatranscriptomics analysis (Supplementary Data 3). From PCA analysis, no major clustering was observed, although post flight samples were located closer to one another in the PCA space (Fig. 4a). The shifts of the samples were mostly from species from *Staphylococcus* and *Streptococcus* family, along the PC2 axis. Slight decrease in overall numbers of bacterial and viral species was observed in postflight samples relative to preflight, with one exception of C003 in metagenomics data and of C004 in metatranscriptomics data (Fig. 4b). Gross comparison of bacterial species by family showed decreased abundance in *Actinobacteria* (e.g., *Actinomyces sp000220835*) while

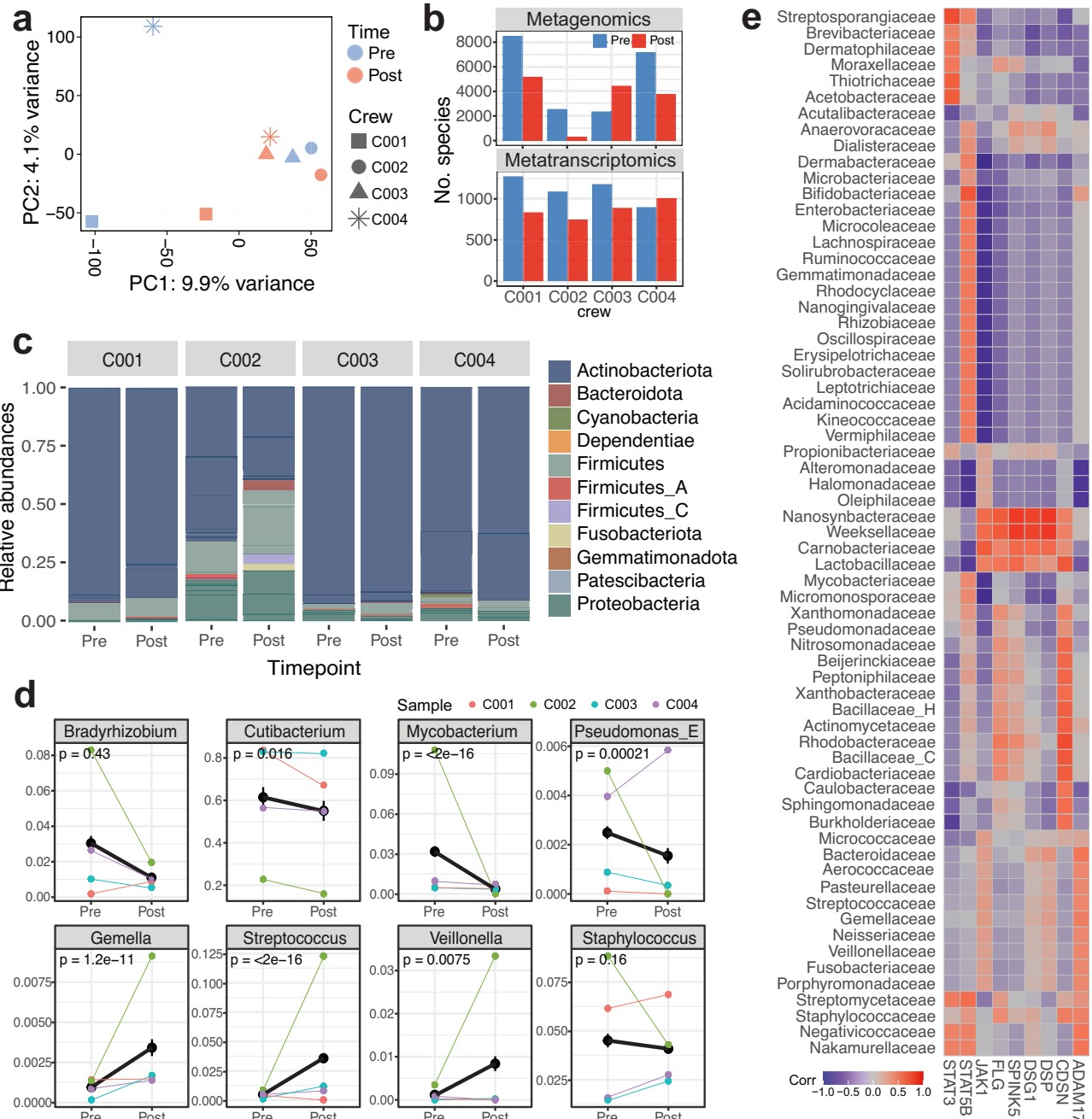

**Fig. 4 | Skin metagenomics and transcriptomics analysis for skin-microbiota interaction map. a** PCA across all metagenomic and metatranscriptomic (bacterial and viral reads) relative abundance features and all crew members pre- and post-flight, **b** Total number of bacterial and viral species with nonzero counts, **c** Relative abundances by sample and timepoint, grouped by family, **d** Changes in relative abundance before and after spaceflight, grouped by genus; statistically significant or previously reported microbes are visualized (two-sided Wilcoxon test across four crew members was performed to compare means between pre- and post-flight samples and to obtain $p$ values, and error bars represent the standard error of the mean), and **e** Correlation across relative abundance of bacterial phyla identified by metagenomics data and known barrier/immune genes associated with skin diseases and disruptions. Source data are provided as a Source Data file.

increased abundance in *Firmicutes/Bacillota* (e.g., *Peptoniphilus C/B*) and *Proteobacteria/Pseudomonadota* (e.g., *Caulobacter vibrioides, Sphingomonas carotinifaciens, Roseomonas mucosa/nepalensis*) (Fig. 4c, d and Supplementary Fig. 6b). When grouped into genus, several species, including *Cutibacterium* (e.g., *Cutibacterium acnes/avidum/modestum/porci*), *Mycobacterium* (e.g., *Mycobacterium paragordonae, Mycobacterium phocaicum*), and *Pseudomonas* (e.g., *Pseudomonas aeruginosa/nitroreducens*) showed statistically significant decrease ($p$ values < 0.05). Several species including *Streptococcus* (e.g., *Staphylococcus capitis, Streptococcus mitis BB*) and *Veillonella*

(e.g., *Veillonella atypica/parvula/rogosae*) showed significant increase (Fig. 4d). Also, species under the *Staphylococcus* genus, such as *staphylococcus capitis/epidermidis/saprophyticus* showed slight decrease while the relative abundances were highly variable across biological replicates.

Changes of bacterial species were then linked to skin gene expression profiles, especially dermatitis-related genes (i.e., *STAT3, STAT5B, FLG, CDSN,* and *ADAM17*) previously associated with *Staphylococcus* species, as *Staphylococcus aureus*-dependent atopic dermatitis have been reported to activate immune system and reduce

microbe diversity[35–37] (Fig. 4e and Supplementary Fig. 6c). When subsetting previously reported bacterial species and associated genes, we found *Staphylococcus* species show an inverse relationship with *JAK1* (Fig. 4e). In particular, *Staphylococcus* correlates closely to *FLG*, *SPINK5*, and *DSG1*, all of which are related with epithelial barriers (stratum corneum and junctional barriers)[38]. Also, microbes belong to *Carnobacteriaceae*, *Lactobacillaceae*, *Nanosynbacteraceae*, and *Weeksellaceae* families showed high correlation with both barrier and immune genes (*CDSN*, *DSP*, *DSG1*, *SPINK5*, *FLG*, and *JAK1*), whereas common skin microbes from *Dermatophilaceae* and *Dermabacteraceae* families showed no correlation. Although larger sample size is needed, it is possible that skin microbiome disruptions, such as those observed in these bacterial families, also contribute to barrier disruption and immune activation during short-term spaceflight.

In addition, from alignment to known viral assemblies we found statistically significant decrease in abundance of reads associated with those from *Uroviricota* (i.e., *Fromanvirus*, *Acadianvirus*, *Armstrongvirus*, *Amginevirus*, *Bixzunavirus*) and *Naldaviricetes* (i.e., *Alphabaculovirus*), and increased abundance of reads associated with those from *Negarnaviricota* (i.e., *Almendravirus*, *Orthotospovirus*) and *Cossaviricota* (i.e., *Betapapillomavirus*, *Betapolyomavirus*) (*p* values < 0.05). Virome changes are limited by the depth of the sequencing and skin virome knowledge, however we also report relative abundances of both bacterial and viral species (Supplementary Data 3). To explore microbiota-skin interactions, we also identified potential associations between microbiome shifts from metagenomics/metatranscriptomics data and human gene expression from skin spatial transcriptomics data; these included associations were with viral phyla (i.e., *Uroviricota*, *Cressdnaviricota*, *Phixviricota*), which is a potentially interesting area to explore as more crew samples are collected. (Supplementary Fig. 6d, e and Supplementary Data 3).

### Immune changes in response to spaceflight

To investigate immune changes that occur beneath the epidermis we also examined changes in immune cells in the profiled vascular regions vs. PBMCs. We saw overall decrease of T cells and increase of macrophage DCs in VA ROIs (Fig. 3b), indicating an immune-epidermis interaction. Related to this, we also observed increased cytokines and inflammatory signals including *IL4*, *IL5*, and *IFNG* in the inner regions (VA and OD ROIs) of the tissue (Fig. 5a)[39,40]. As a confirmation, we observed that these specific cytokines are also shown to be increased in cytokine assays from the crew members' serum samples (Fig. 5b). To compare immune change observations from VA ROIs to system-wide immune system changes, we performed leveraged 10X multiome sequencing (dual snRNA and ATAC sequencing from each cell) on timepoint-matched PBMCs from the crew members (Supplementary Fig. 7a). We analyzed 151,411 cells across 9 gross cell types and performed differential expression analysis (Supplementary Fig. 7b, c). Overall, we observed fluctuations of T-cells across timepoints, consistent to the observations from skin spatial transcriptomics data (Fig. 5c, d and Supplementary Fig. 7d). Among 555 DEGs from multiome samples and 446 DEGs from GeoMx VA ROIs, 12 overlapping DEGs were found (both log2FC > 0.1 and *p* values < 0.01, DESeq2), including *ATP11A*, *CEP85L*, *CEPT1*, *DMXL1*, *DOP1A*, *EVI5*, *GSAP*, *MDFIC*, *SENP7*, *TBCK*, *VAV3*, and *VPS13C* (Fig. 5c and Supplementary Fig. 7c). Several of these genes are related to cellular metabolism and cytosolic transports. In particular, *VAV3*, one of signaling adapters in NK/T cell activation, has been previously reported to be associated with atopic dermatitis onset[41–43]. While all these overlapping DEGs were temporary in PBMCs, i.e., upregulated in the immediate postflight samples (R + 1 timepoint) and returned to pre-flight expression levels, the chromatin accessibility of these genes stayed slightly longer, up to R + 45 timepoint (Fig. 5d).

Finally, we derived cell type- and spaceflight-specific gene signatures from the multiome data, to examine any enrichment in the GeoMx samples (using single-sample gene set enrichment analysis, or ssGSEA approach) (Fig. 5e). Most of the immune cell specific postflight DEGs enrichments were near the innermost ROIs (OD and VA), except for T cells (both CD4+ and CD8+), which showed enrichment in the postflight OE ROIs. While it was previously reported that spaceflight stressors change the immune system, increased enrichment of the T cells in the epidermal region correlates with activated T cell activity and connects to inflammatory responses and barrier disruptions[44–48]. Lastly, we found that these increased T cell signatures in the OE region may not have direct connection to Th17 T cells or psoriasis, rather have closer connection to the antigen-associated and lymphatic T cells infiltrated from inner layers of the skin (Fig. 5f)[49,50]. Also, the ssGSEA analysis using skin disease-associated gene signatures showed a slight increase in melanoma signatures. The slight increase can be explained with previous observations throughout this manuscript, including increase in cell death, immune activation, and stress response (Supplementary Fig. 7e, f), but more research is needed to prove the direct connection or causality of gene expression shifts.

## Discussion

In this study, we applied a battery of omics methods of the Inspiration4 crew to discern the impact of spaceflight on the skin, collecting and integrating skin spatial transcriptomics data with clinical, metagenomic, metatranscriptomic, and single-cell sequencing data. By performing spatially resolved sequencing of astronaut skin biopsies before and after short-term spaceflight, we see varying transcriptomic and cellular changes and responses specific with their four skin layers (outer and inner epidermis, dermis, and vasculature) across 95 ROIs from all four crew members. To provide a more comprehensive view of these changes in response to the spaceflight environment and stressors, we correlated the spatial expression profiles with the metagenomics and metatranscriptomics data with the skin swab and PBMC single-cell sequencing data collected from the same crew member and from the same biopsy locations.

The Inspiration4 mission was a short-term (3-day flight) spaceflight, with minimal exposure to radiation (about 2.7 mGy-Eq based on NASA Q quality factors and 50% FAX ModelEffective dose). Most of these changes are expected to be temporary, based on prior missions' data, and due to fluid redistribution and stress from the spaceflight itself. At the molecular level, we observed consistent changes in immune signals and extracellular matrix and junction proteins in the postflight samples relative to preflight. Most evidently, we see a strong increase in gene expression of the KRAS pathway, which has previously been discussed in the context of space radiation[51]. In the context of skin, KRAS is known to alter epidermis homeostasis and induces redundant skin, papillomas, defective skin cell proliferation and differentiation and associated with many skin diseases such as melorheostosis, lymphangiomatosis, and vascular stenosis[52–54]. Activated KRAS, as a part of RAS/MAPK signaling pathway, is responsible for many angiogenic pathways, cytokine/chemokine productions, biochemical pathways (e.g., *PI3K*, *RAF*, etc.), and cellular proliferation/renewals[55]. When activated, the KRAS oncogene has the greatest mutation rate of all cancers, and is linked to several difficult to treat, highly lethal cancers, such as pancreatic ductal adenocarcinoma, non-small-cell lung cancer, and colorectal cancer[56].

In the context of keratinocyte biology, KRAS plays various roles including proliferation, survival, cell-cell adhesion, and polarity and redox balance, and the Ras pathway is also activated by integrins and negatively regulated by cell-cell adhesion[57]. KRAS has many roles across three key cell types, including: (1) endothelial cells (i.e., mechanotransduction pathways and integrin-associated events[58]), (2) fibroblast reprogramming (i.e., inflammatory gene expression and tissue extracellular signaling effectors, which polarizes immune populations while preventing tissue repair[59]), and (3) pericytes (i.e., wound healing and scar formation, and endothelial tubulogenesis, plus

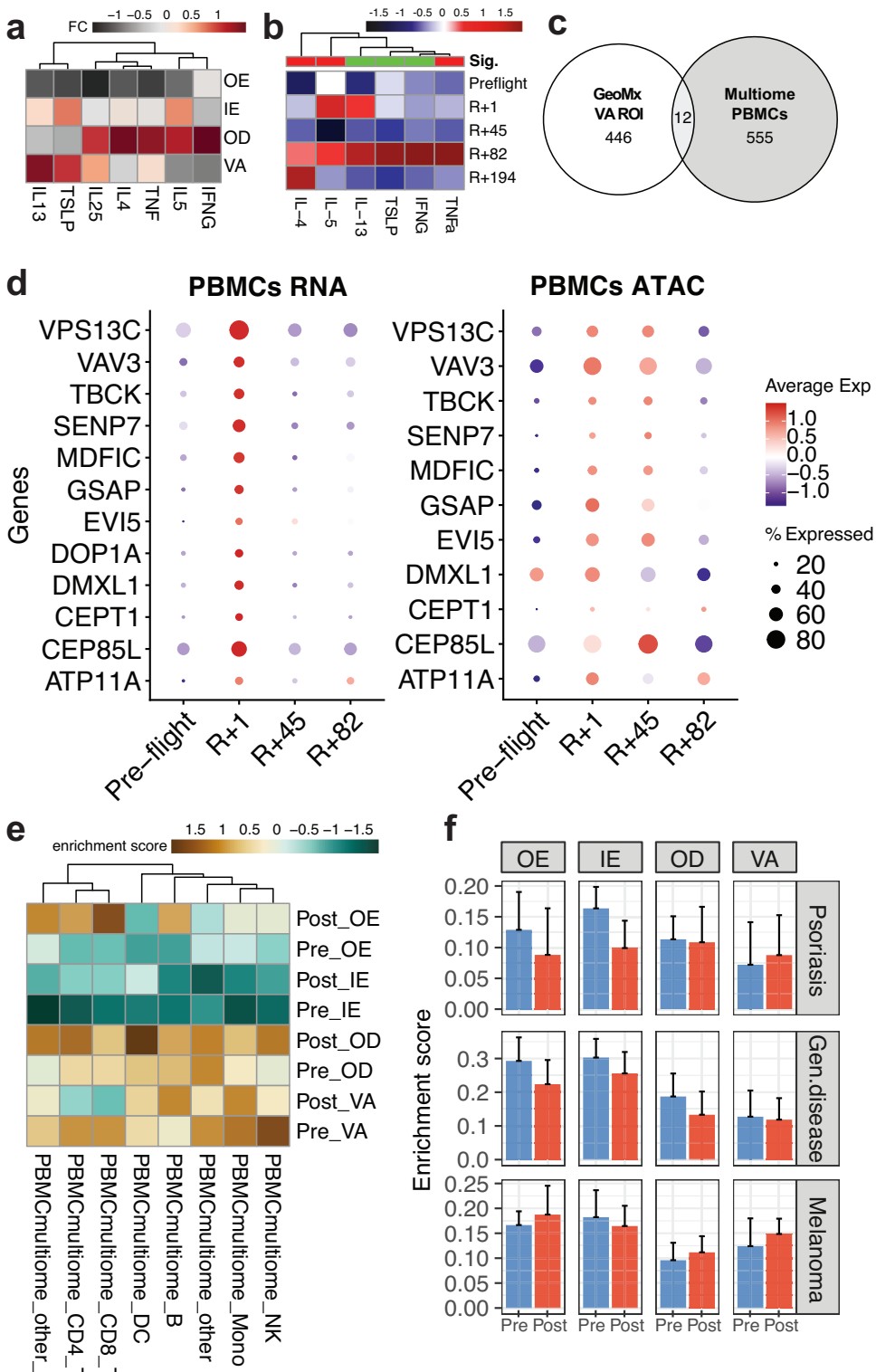

**Fig. 5 | Changes in the skin immune system relates to peripheral blood immune cell changes. a** Notable cytokine changes and locations from (**a**) skin transcriptomics data by region and, **b** cytokine assay from serum samples (sig. Indicates overall statistical significance of the cytokine levels in the postflight samples relative to the preflight samples, where red indicates significantly increased, and green means stable/no change; two-sided Wilcoxon test was done with the *p* value cutoff of 0.05), **c** Comparison of DEGs between PBMC multiome data and spatial transcriptomics data from VA ROIs, **d** Dot plots visualizing mRNA transcript expression levels (left) and gene activity score from ATAC signals (right), where preflight samples were collected 44 days before launch (L-44) and postflight samples were collected 1, 45, and 82 days post return (R + 1, R + 45, and R + 82, respectively), **e** Flight and cell type specific gene signature enrichment in spatial data by timepoint and ROI types, **f** gene signature enrichment analysis using gene signatures built from skin disease-related gene expression profiles; two-sided Wilcoxon test across four crew members and 95 ROIs was performed to obtain *p* value, where *\*p ≤ 0.05* and *\*\*p ≤ 0.01*, and error bars represents standard deviation of the mean. Source data are provided as a Source Data file.

apical membrane and cytoskeletal polarization, which mediates the interaction between endothelial cells and pericytes[60,61]). From our results, we also see changes in various cell populations, including melanocyte, pericyte, fibroblast, and lymphatic endothelial cells, with KRAS being a likely mechanistic pathway being altered from spaceflight exposure within these cell populations leading to their phenotype changes. The impairment of skin cell proliferation is consistent with relative decrease in melanocytes and fibroblasts in postflight samples.

Related to this observation, other non-inflammatory pathways such as WNT-βcatenin, TGFβ, Notch, ROS, glycolysis, apoptosis, oxidative phosphorylation, G2M checkpoints, also show significant changes. These pathways may provide more context as to why KRAS is changing, as well as the phenotypic adaptations and distribution changes seen with the cell populations. For example, from a metabolic perspective, an increase in glycolysis and a decrease in oxidative phosphorylation is tied with apoptotic events, decreased oxygen supply/hypoxia (e.g., change in cardiovascular function, increased inflammation, increased wound healing, etc.), as a compensatory response to the environmental insult. TGFβ is involved with ECM remodeling, inflammatory pathways, and these various cell types affect these biological events. Notch and WNT-βcatenin are involved with cytoskeletal/cell-cell adhesion processes. Furthermore, looking at the host of inflammatory pathways changing (interferon, TNFα, complement, interleukin), primarily pro-inflammatory given the timeframe of this mission as well as environmental insults (e.g., radiation exposure, weightlessness) that also exacerbate the biochemical pathways that are being shifted after spaceflight (e.g., metabolic, cell phenotypic, increased apoptosis, etc.).

The spatial analysis enabled a unique study of gene expression changes, at higher resolution across the layers of skin, plus also within distinct tissue compartments. While all regions show varying but consistent enrichment in inflammation-related pathways, we are able to locate damage and repair related pathways (i.e., DNA repair, apoptosis, and UV response, reactive oxygen species) in epidermal (mostly outer layers) regions, suggesting that the skin's stressors from spaceflight occur primarily in the outermost region. Space radiation can cause damage to DNA directly through interaction of charged particles with DNA itself or indirectly through the production of free radicals (which we also see from the increased oxidative stress transcripts)[62]. The type of DNA damage can also differ, from single-strand to double-strand breaks, chromosome aberrations, copy number variation, and other genomic changes. The level of DNA repair depends on the cell type and its capacity for repair as well, with more rapidly replicating cell types (e.g., epithelial cells vs. endothelial cells) having more active DNA repair mechanisms[63]. Of note, spaceflight can also cause increased reactive oxygen species production, likely also from radiation exposure[64–68]. We also identify specific keratin (KRT) transcripts in dermal layer and loss of fibroblast and cell junction transcripts including DES, ACTA2, TLN1, and TAGLN in vascular regions, which may be related to epithelial barrier disruptions and hindered regeneration. Various spaceflight and ground-analog studies have shown spaceflight induces oxidative damage to cell junctional proteins and adhesion.

Further supporting epithelial barrier disruptions, we show decrease in barrier-related transcripts (i.e., FLG, HAS, OCLN, CLDN) and relate the loss of these proteins to microbiome changes using metagenomics and metatranscriptomics data from the swabs. Some of these changes may be related to the changes in skin microbiota. Consistent to previous reports from the International Space Station (ISS), we find changes in microbes such as Corynebacterium, Actinomyces, and Staphylococcus in the Inspiration4 crew skin samples, despite the small sample size[69]. We then correlated microbiome changes with gene expression fluctuations and identified gene-microbe clusters. While many of these changes are known to be associated with the microbial composition within the capsule,

systematic studies spanning spaceflight environment, skin surface, and gene expression will provide additional insights to which countermeasures are needed[70,71].

Connecting the findings in the immune system, we cross-validated these results with 10X multiome profiling of PBMCs and reported consistent immunological changes. We identify overall loss of immune cells in vascular ROIs, while T-cell signature increases in the outermost epidermal layers. T-cell migration and infiltration in epidermal layers have been previously associated with skin inflammation and several disease conditions[72], and several other studies have connected immune system dysregulation and adaptations to spaceflight exposure. For example, consistent adaptive immune system changes (for example, T cells generally shift toward a Th2 phenotype) are consistent with our observation that T cells shifts can lead to fluctuations in cytokine concentration[44,73]. We also found increases in macrophages and various chemotactic markers (e.g., CCL2, CCL4, CXCL5), which are previously associated with spaceflight. These changes, which were predicted in peripheral, circulating blood cells, are shown within the vasculatures of the skin tissue microenvironment. Furthermore, by taking intersections of DEGs from both skin spatial profiling and PBMC single-cell, we saw an enrichment of genes related to neutrophils. Such observation is consistent with previous observations from space shuttle flights studies[74]. It has been shown that neutrophils increase in postflight samples, likely due to inflight stress rather than microgravity.

Related to this observation, the changes in the microvascular network seen (both blood and lymphatic), may also be suggestive of local tissue adaptations resulting from fluid redistribution and interstitial pressure changes. It has been previously shown that microgravity-induced capillary transmural pressure (blood to tissue) change may reduce lymphatic flow and is discussed in the context of ocular health[8,75]. These changes in lymphatic flow and cellular microenvironment are also crucial for immune and stem cell interactions and turnovers, therefore deeper investigations to characterize the changes and develop countermeasures are crucial[76,77].

Overall, we present a thorough model of skin tissue (1) interacting with microbes and viruses on the surface, (2) changing expression profiles layer-by-layer, and (3) signaling from circulating PBMCs that likely interact with the peripheral skin vasculature. Our study is limited by four crew members, two timepoints, and a short-term mission, but to mediate this limitation, the data has been compared with other research and analog models (Henry Cope and Jonas Elsborg et al., Nature Communications, in review). In addition, some limitations are due to the unique challenges associated with the astronaut samples, including: their difficult procurement (biopsy), their rarity (there are few missions being conducted), and limited infrastructure for on-site sample collection and processing[78–83]. Nonetheless, these first structural and molecular views into the epidermal response to spaceflight highlight key KRAS, immune, and related pathways that can be integrated in the future missions and on-site monitoring of crews[84] to help researchers understand the long-term effects of the spaceflight stressors and cellular changes we identified in this study.

## Methods

### IRB statement
The research in this manuscript complies with all relevant ethical regulations. Tissue samples were provided by SpaceX Inspiration4 crew members (n = 4, two males and two females) after consent for research use of the biopsies, swabs, and biological materials. The procedure followed guidelines set by Health Insurance Portability and Accountability Act (HIPAA) and operated under Institutional Review Board (IRB) approved protocols. Experiments were conducted in accordance with local regulations and with the approval of the IRB at the Weill Cornell Medicine (IRB #21-05023569).

## Sample collection

**Skin biopsy and swabs.** The biopsies were collected from all four crew members' right (preflight) and left (postflight) deltoid (or upper arm) that were anatomically the same, matched locations, except for one crew member. For this crew member, both timepoints were collected from the right arm, approximately 1.5 inches away from one another. The average age of the crew members was 41.5 and consisted of two males and two females. Prior to biopsy, the selected areas were swabbed for 30 s with pressure using an Isohelix Swab dampened with DNase and RNase free water. The swabs were stored in the pre-labeled Thermo Fisher 2D barcoded tubes with DNA/RNA Shield Stabilization Solution (Zymo Research, cat# R1100-250) added. After the swab, the area was disinfected using CloraPrep (BD Biosciences) followed by injection of 1% lidocaine and 1/100,000 epinephrine. The biopsies were performed using a trephine punch (Miltex). If desired, the surgical sites were either closed with two 5-0 Vicryl sutures or applied pressure to stop bleeding. To prevent infection and allow healing, bacitracin ointment or sterile petrolatum was applied to the biopsied site followed by Band-Aids and after-care instructions. The resected piece was cut into approximately 1/3 and 2/3 sections. The smaller piece was fixed with formalin and stored at room temperature for histology and pathology. The larger piece was flash frozen for GeoMx analysis.

**PBMC isolations.** For each crew member, 8 ml of venous blood was collected in Ethylenediaminetetraacetic acid (EDTA) anticoagulant tubes. Depletion of granulocytes was performed either directly from whole blood using the RosetteSep™ granulocyte depletion cocktail or by cell sorting after PBMC isolation. Whole blood was incubated in a granulocyte depletion cocktail (50 µl/ml of blood) for 20 min at room temperature. Next, Ficoll-Paque Plus (Cytiva) was utilized to isolate PBMCs by density gradient centrifugation. After washes in PBS with 2% FBS (GIBCO) were completed, isolated PBMCs were cell sorted to remove granulocytes only if the RosetteSep™ granulocyte depletion cocktail was not added to whole blood prior to density gradient centrifugation. Granulocytes were identified using side scatter and the lymphocyte and monocyte fractions were sorted using a 100 µm nozzle (BD Aria). Following granulocyte depletion, PBMCs were split into two fractions to generate single-cell V(D)J T-cell and B-cell libraries or Multiome (GEX and ATAC) libraries.

## Spatial transcriptomics analysis

**GeoMx digital spatial profiling.** Skin biopsy samples from four crew members were collected and frozen in cryovials pre and post flight. Collected skin was flash embedded in OCT blocks. Three or four replicate of OCT-embedded tissues were placed on a single slide per subject including pre and post flight replicates. Tissues were then cryosectioned at 5 µm thickness and attached to glass microscope slides (Fisher Scientific, cat# 22-037-246).

Immunofluorescent visualization marker for Pan-Cytokeratin (PanCK, Novus cat# NBP2-33200AF532, Alexa Fluor® 532, clone ID AE1 + AE3, 1:40 dilution), fibroblast activation protein (FAP, Abcam cat# ab222924, clone ID EPR20021, conjugated to Alexa Fluor® 594 using Thermo Fisher antibody labeling kit cat# A20185, 1:20 dilution) and smooth muscle actin (SMA, R&D Systems cat# IC1420R, Alexa Fluor® 647, clone 1A4, 1:200 dilution) were used for region or interest (ROI) selection. The DSP whole transcriptome assay (WTA) was used to assess genes collected in each ROI. For DSP processing, OCT slides were thawed overnight in 10% neutral-buffered formalin (NBF) at 4 °C followed by PBS washes for thorough fixation. After washes, slides were prepared following the automated Leica Bond RNA Slide Preparation Protocol for fixed frozen samples, digesting samples with 1.0 µg/ml proteinase K for 15 min, and antigen retrieval for 20 min at 100 °C (NanoString, no. MAN-10115-05). In situ hybridizations with the GeoMx Whole Transcriptome Atlas Panel (WTA, 18,677 genes) at 4 nM final concentration were done in Buffer R (NanoString). Morphology markers were prepared for four slides concurrently using Syto13 (DNA), PanCK, FAP and SMA in Buffer W for a total volume of 225 µl per slide. Slides incubated with 225 µl of morphology marker solution at RT for 1 h, then washed in SSC and loaded onto the NanoString DSP instrument. The 20x scan was used to select freeform ROIs to guide selection of outer epidermal (OE), inner epidermal (IE), outer dermal (OD) and vascular (VA) regions. OE ROIs covered spinous and granular layers, while IE ROIs covered the basal layer, identified from the staining of the tissue. To ensure proper selection and to avoid overlaps across different ROI types, small gaps between each ROI type were made, as shown in Supplementary Fig. 1.

GeoMx WTA sequencing reads from NovaSeq6000 were compiled into FASTQ files corresponding to each ROI. FASTQ files were then converted to digital count conversion files using the NanoString GeoMx NGS DnD Pipeline. From the normalized count matrix, DESeq2 was used to perform differential expression analysis (Supplementary Data 1), and FGSEA was used for pathway enrichment analysis, which uses Kolmogorov-Smirnov-like enrichment score test to obtain statistics (Supplementary Data 2). GSVA was used to run ssGSEA analysis with custom gene signatures obtained from published single-cell and bulk skin datasets[25,26]. The methods used to perform data exploration and visualization have been published previously[78].

## PBMC single-cell sequencing analysis

**Library preparation and sequencing.** To capture T cell and B cell variability, diversity, and joining (VDJ) repertoire, single-cell gel beads-in-emulsion and libraries were performed according to the manufacturer's instructions (Chromium Next GEM Single Cell 5′ v2, 10x Genomics). Prior to single-cell Multiome ATAC and gene expression sequencing, nuclei isolation was performed by resuspending PBMCs in 100 µl of cold lysis buffer containing 10 mM Tris-HCl (pH 7.4), 10 mM NaCl, 3 mM MgCl$_2$, 0.1% Tween-20, 0.1% Nonidet P40, 0.01% digitonin, 1% BSA, 1 mM DTT and 1 U/µl RNAse inhibitor. Cells were incubated for 4 min on ice, followed by the addition of 1 ml cold wash buffer (10 mM Tris-HCl (pH 7.4), 10 mM NaCl, 3 mM MgCl$_2$, 0.1% Tween-20, 1% BSA, 0.1% Tween-20, 1 mM DTT, 1 U/µl RNAse inhibitor). After centrifugation (1000 × $g$ for 5 min at 4 °C), nuclei were resuspended in diluted nuclei buffer (10x Genomics Single-Cell Multiome ATAC kit A) at a concentration of 6000 nuclei per µl. Single-cell libraries were generated via the Chromium Next GEM Single-Cell Multiome ATAC and Gene Expression kit (10X Genomics) according to the manufacturer's instructions. The libraries were sequenced on the NovaSeq 6000 sequencing system. GEX and ATAC libraries were processed using Cell Ranger arc v2.0.0. VDJ libraries were processed using Cell Ranger v6.1.1. Reads were aligned to the GRCh38 human genome.

## Metagenomics and metatranscriptomics analysis

**DNA/RNA library preparation and sequencing.** Samples were collected prior to processing as isohelix swabs stored in 400 ul of Zymo Research DNA/RNA shield (cat# R1100) in Thermo Fisher Matrix 1.0 ml ScrewTop Tubes (cat# 3741-WP1D-BR). Prior to beginning the extraction protocol, samples were vortexed for 5 s using the MO BIO Vortex Adapter tube holder (cat# 13000-V1-24) to ensure maximum biomolecular yield from the collected swabs. Due to loss in DNA/RNA shield during sample collection, sample volume was transferred into AllPrep Bacterial Bead Tubes provided by the Qiagen Allprep Bacterial DNA/RNA/Protein Kit (cat# 47054) and the volume of samples were adjusted to 350 ul prior to further processing to ensure sufficient input volume for the extraction protocol. DNA/RNA/Protein extraction was performed using the Qiagen Allprep Bacterial DNA/RNA/Protein Kit (cat# 47054). Steps one and two were omitted from the Qiagen Allprep Bacterial DNA/RNA/Protein Kit manufacturer's protocol as these samples were not generated through bacterial culture. Aside from this omission, samples were extracted following the manufacturer's protocol. Following the completion of the extraction protocol, the

extracted DNA was quantified using Thermo Fisher Qubit 1X dsDNA HS Assay (cat# Q33231), extracted RNA was quantified using Thermo Fisher Qubit RNA HS Assay Kit (cat# Q32855), and extracted protein was measured using the Thermo Fisher Rapid Gold BCA Protein Assay (cat # A53225). All quantification protocols were conducted per manufacturer's instructions.

Illumina DNA Library prep kit was used for preparing all extracted DNA and cDNA (from RNA) samples for Illumina Whole genome sequencing per manufacturer's instructions. Adapters used were IDT® for Illumina® DNA/RNA UD Indexes, Tagmentation (96 Indexes, 96 Samples). In total, 408 DNA samples (4 × 96-well plates and 24 samples on a 5th 96-well plate) were library prepared and pooled into a total of 4 pools—96 samples were pooled into 1 (for plates 1, 2 and 3), except for plates 4 and 5, where 120 libraries were pooled into 1. Each library pool was sequenced using 2 × 150 Paired end sequencing on S4 flow cell using NovaSeq 6000 sequencer, one pool per lane of the S4 flow cell.

**Genomic data preprocessing and quality control.** All metagenomic and metatranscriptomic samples underwent the same quality control pipeline prior to downstream analysis. Software used was run with the default settings unless otherwise specified. The majority of our quality control pipeline makes use of bbtools (V38.92), starting with clumpify [parameters: optical = f, dupesubs = 2, dedupe = t] to group reads, bbduk [parameters: qout = 33 trd = t hdist = 1 k = 27 ktrim = "r" mink = 8 overwrite = true trimq = 10 qtrim = "rl" threads = 10 minlength = 51 maxns = −1 minbasefrequency = 0.05 ecco = f] to remove adapter contamination, and tadpole [parameters: mode = correct, ecc = t, ecco = t] to remove sequencing error[79]. Unmatching reads were removed using bbtool's repair function. Alignment to the human genome with Bowtie2 (parameters: --very-sensitive-local) was done to remove potentially human-contaminating reads[80].

**Taxonomic abundance quantification.** We used the XTree (https://github.com/GabeAl/UTree) [parameters: –redistribute] to quantify the bacterial and viral taxonomic compositions of the metagenomic and metatranscriptomic samples. XTree is the recent version of Utree[81]; it implements an optimized alignment approach and increased ease of use. In brief, XTree is a k-mer based aligner (akin to Kraken2[82], but faster and designed for larger databases) that uses capitalist read redistribution in order to pick the highest-likelihood mapping between a read and a given reference based on the overall support of all reads in a sample for said reference[83]. Notably, it reports the total coverage of a given query genome, as well as total unique coverage, which refers to coverage of regions found in only one genome of an entire genome database.

We generated an XTree k-mer database [parameters: XTree BUILD k 29 comp 0] from the Genome Taxonomy Database representative species dataset (Release 207) and aligned both metagenomic and metatranscriptomic samples. For the metagenomic sequencing, we filtered for genomes that had at least 0.5% coverage (i.e., approximately 15 genes in 3000 genes, each 1000 bp) and/or 0.25% unique coverage. Relative abundances were calculated by dividing the total reads assigned to a given genome by the total number of reads assigned to all genomes in each sample.

**Cytokine assay**
Biochemical assays from the serum samples obtained from the same i4 crew members was performed using the bead-based multiplexed assay protocols provided by Eve Technologies (https://www.evetechnologies.com/cytokine/). Two biomarker profiling panels were used: (1) Human Cytokine/Chemokine 71-Plex Discovery Assay® Array (HD71) and (2) Human Cardiovascular Disease Panel 3 9-Plex Discovery Assay® Array (HDCVD9). Concentration values from the assay were extrapolated using a 4 of 5 parameter logic standard curve.

**Chromogenic in situ hybridization**
Using additional sections from the same FFPE blocks, AP3B1 transcripts were quantified using custom RNA probe. Five-micron thick sections were cut from the paraffin blocks and mounted on glass slides. For each block two duplicate serial sections were mounted on a single slide, stained, and analyzed. Chromogenic in situ hybridization was performed on an automated stainer (Leica Bond RX, Leica Biosystems, Deer Park, IL) with RNAscope 2.5 LS Assay Reagent Kit-Red (Cat. # 322150, Advanced Cell Diagnostics, Newark, CA) and Bond Polymer Refine Red Detection (Cat. # DS9390, Leica Biosystems) following manufacturer's standard protocol. A probe was designed to detect region 301-1333 of human AP3B1 gene, NCBI Reference Sequence 126221 (Cat. # 1262218-C1; Advanced Cell Diagnostics). Positive control probe detecting a housekeeping gene (human PPIB, cat # 313908, Advanced Cell Diagnostics) and a negative control probe detecting bacterial (Bacillus subtilis) dapB gene (cat #312038, Advanced Cell Diagnostics) were used to confirm adequate RNA preservation and detection, and absence of non-specific signal, respectively. The chromogen was fast red and the counterstain hematoxylin. Positive RNA hybridization was identified as discrete, punctate chromogenic red dots under brightfield microscopy. Images were acquired with an Olympus VS200 slides scanner and VS200 ASW 3.4.1 software (Evident Scientific, Hamburg, Germany) using a 40X/0.95NA objective and extended focal imaging mode to generate whole slide images with a pixel size of 0.1369 μm. Images were analyzed using Halo 3.5.3577 software and ISH module 4.2.3 (Indica Labs, Albuquerque, NM). The entire epidermis region was annotated manually and analyzed. As melanin pigment was observed in the epidermis it was set as an exclusion stain in order to avoid its false detection as red stain. Probe size was set at 1.0000 μm². Average probe copy number per cell was reported. Image acquisition and analysis were performed by a board-certified veterinary pathologist.

**Reporting summary**
Further information on research design is available in the Nature Portfolio Reporting Summary linked to this article.

## Data availability
All the raw sequencing data (skin spatial transcriptomics, swab metagenomics and metatranscriptomics) as well as images, processed data, and associated metadata are submitted to NASA GeneLab data sharing platform with submission ID OSD-574. PBMC single-cell multiome data are also available via NASA GeneLab platform under OSD-570. Individual-level data are only identifiable with IRB, however, aggregated data and interactive online visualization tools are available at: https://soma.weill.cornell.edu/apps/SOMA_Browser/. Source data are provided with this paper.

## Code availability
Codes used to analyze skin spatial transcriptomics, processed data used to generated figures, and skin-specific custom gene sets are accessible in GitHub repository: https://github.com/jpark-lab/SpatialAnalysis/[85]. All other data analyses including single-cell multiome and metagenomics data analysis can be found in https://github.com/eliah-o/inspiration4-omics.

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

## Acknowledgements

We thank Igor Tulchinsky and WorldQuant, The GI Research Foundation, NASA (NNX14AH50G, NNX17AB26G, 80NSSC22K0254, NNH18ZTT001N-FG2, 80NSSC22K0254), the National Institutes of Health (R01MH117406, P01CA214274 R01CA249054), and the LLS (MCL7001-18, LLS 9238-16, 7029-23). We also would like to acknowledge the Laboratory of Comparative Pathology (supported by Cancer Center Support Grant P30 CA008748) for histology and RNA scope analysis. Additionally, the authors would like to thank following grants and/or fellowships: Bumrungrad International Hospital (J.P.), MOGAM Science Foundation (J.P., J.K.), NASA Space Biology Postdoctoral Fellowship (80NSSC19K0426, S.A.N.), Human Research Program Augmentation Award (80NSSC19K1322, S.A.N.), and the Boryung Care in Space (CIS) Award.

## Author contributions

C.E.M., E.G.O., and J.P. conceived and designed the experiments. GeoMx data were processed by A.R., B.M.H., and S.E.C. with the help from J.P., E.G.O., N.D., D.N., K.R., J.Pr., A.K., J.G., and J.W.H. Spatial transcriptomic data analysis and statistical investigation was done by J.P. with B.M.H. and S.E.C. R.G. and E.A. processed and analyzed tissue samples with S.M. and J.M. Data analysis, interpretation, and figures

were made by J.P. with the inputs from J.K., S.N., and B.T. Initial manuscript was written by J.P. with the help from J.K., S.N., B.T., N.D., D.N., E.G.O., C.M., J.Pr., A.K., J.W.H., G.C., A.B., M.M., and C.E.M. All authors discussed the results and contributed to the final manuscript.

## Competing interests

C.E.M. is a scientific advisor of NanoString Inc. A.R., B.M.H., and S.E.C. are employees of NanoString Technologies, and J.M. and S.M. are employees of SpaceX. The remaining authors declare no competing or relevant interests.

## Additional information

[1]Department of Physiology, Biophysics and Systems Biology, Weill Cornell Medicine, New York, NY, USA. [2]The HRH Prince Alwaleed Bin Talal Bin Abdulaziz Alsaud Institute for Computational Biomedicine, Weill Cornell Medicine, New York, NY, USA. [3]Department of Nutrition & Integrative Physiology, Florida State University, Tallahassee, FL, USA. [4]Department of Dermatology, Weill Cornell Medicine, New York, NY, USA. [5]NanoString Technologies, Inc., Seattle, WA, USA. [6]SpaceX, Hawthorne, CA, USA. [7]Department of Genetics, Harvard Medical School, Boston, MA 02115, USA. [8]Wyss Institute for Biologically Inspired Engineering, Harvard University, Boston, MA, USA. [9]Stanley Center for Psychiatric Research, Broad Institute of MIT and Harvard, Cambridge, MA, USA. [10]Blue Marble Space Institute of Science, Space Biosciences Division, NASA Ames Research Center, Moffett Field, CA, USA. [11]The Feil Family Brain and Mind Research Institute, Weill Cornell Medicine, New York, NY, USA. ✉e-mail: chm2042@med.cornell.edu

