## [Peer Review File · Nature Communications]

REVIEWER COMMENTS

Reviewer #1 (Remarks to the Author):

In this manuscript Park and co-authors employ multiomics methods to analyze skin and blood samples from space mission crew members pre- and post- spaceflight to map the spatial transcriptome in healthy skin with paired metagenomics metatranscriptomics of skin swabs and sc-RNAseq of PBMCs. They demonstrate differences in cell types abundance and inflammatory and KRAS signaling activation in response to spaceflight.

The study is underpowered and primarily descriptive, with little biological insights emerging and no experimental validation of the claims *in vitro/in vivo/ex vivo*. It also appears by the claims of this paper that the authors do not have a deep understanding of fundamental cutaneous biology concepts.

Major concerns:

- An important paper from the space biology field also using multi-omics analyses to characterize spaceflight impact is not referenced at all (PMID: 33242417). How do the results of da Silveira et. al compare to the findings of this manuscript?
- KRAS involvement has previously been shown in spaceflight (PMID: 32045996); also not referenced by the authors.
- Can the authors explain the time point selection? why 44 days before the mission and then 1 day after?
- Were biopsies taken from sun-protected regions? Same arms? How close were the pre- to post-spaceflight ones. Age/sex of spaceflight crew?
- Since freeform ROI selection was completed and it appears from Fig 1a that quite large regions were selected (e.g. for the dermis and epidermis), were all ROIs from the same patient and anatomical location distinct or was there overlap and how did the authors ensure no overlap? Could the authors show for a representative study subject all the ROIs selected?
- how was batch effect corrected?
- The authors in multiple instances report genes and cell types that are impossible to find in the anatomical locations mentioned. Line 136 "Loss of keratin family transcripts... found in the dermal layer"? could this be explained by a flaw in the selection of ROIs where epidermal regions were included in the pre-flight samples? no keratins are normally expected in the dermis. Line 157 "loss of melanocytes in the OD" Melanocytes are only present in the basal epidermis. Line 160 "fibroblasts in the OE" no fibroblasts are present in the epidermis.

Line 178, reticular fibroblasts are not differentiated fibroblasts

Fig. 3b showing lymphatic EC, pericytes and fibroblasts in the OE, or melanocytes in the VA is incorrect.

Also Extended Fig. 3a shows proportions of cells that are impossible to find in the described compartments. Fibroblasts even in the outer epidermis? Melanocytes throughout the skin?

Line 224 “In the OE... increased lymphatic EC and decrease in endothelial cells”. There are no vessels in the epidermis, especially the outer epidermis.

All these claims seriously compromise the biological validity of the study.

- ATAC-seq is mentioned but not shown?

- Line 333, what do you mean by “skin cells” ? endothelial cells are also skin cells

- Line 442 “custom gene signatures obtained..” can the authors provide these signatures, or at least reference the studies that used them ?

Minor comments:

- Please clarify (L-44) and (R+1) terminology for readers unfamiliar with space travel terms

-Line 421, type of slide used? Also how thick were the cryosections?

Line 252, please correct to “atoPic dermatitis”

Reviewer #2 (Remarks to the Author):

The manuscript by Park et al. titled “Spatial multi-omics of human skin reveals KRAS and inflammatory responses to spaceflight” aims to describe the changes that spaceflight causes to the skin (different sites), using GeoMx™ Digital Spatial Profiler and multi omics assays (single cell RNA-seq, metagenomics, and metatranscriptomics). The authors describe the increase in inflammatory genes and changes in other immune effectors post-spaceflight. In addition, the authors describe the microbial changes in the skin. While there are important implications to the observations the descriptive nature of the results and the lack of validation experiments require additional experimentation and prevent the publication of the manuscript in its current form.

Comments:

The authors should look for changes in antimicrobial resistance in the skin microbiome using the metagenomic and transcriptomic data.

The authors should also present the virome changes due to spaceflight.

The authors should discuss the limitations of the study small sample size

For the pathology shown please add a pathology score, so the reader without training can assess the changes to the tissue.

In line 140, the authors describe that Ap3b1 transcript was upregulated. The authors should validate this information via qPCR or western blot.

In line 174 the authors state that there was a decrease in protein production, but the data suggest it is gene expression. Please clarify. Also, IHC or IF for protein cell markers would provide additional validation to the data presented and the single cell data.

189: Did the swab was taken in the exact spot of the biopsies?

The authors constantly state that certain species were changed due to spaceflight, but the data shows only microbiome changes at the genus level. If metagenomic data was collected the authors should show at least show the top species.

Metagenomics could be used to assemble DNA sequences for the microbiome to use the transcriptomic data to be mapped to it. Thus a multiomics analysis of the skin microbiome.

The paragraph of lines 220 to 233 should be moved to the discussion. It lacks enough data for the stated conclusions.

Cytokine data has no protein-based or qPCR validation or validation in the tissue acquired.

The authors should integrate the microbiome data with the immune data (figure 5) and assess what microbial changes correlate with inflammation and immune cell changes.

In extended data figure 2, there seem to be only one sample for C002 for OE and IE.

Point-to-point response to reviewers' comments

Reviewer #1

(Remarks to the Author):

In this manuscript Park and co-authors employ multiomics methods to analyze skin and blood samples from space mission crew members pre- and post- spaceflight to map the spatial transcriptome in healthy skin with paired metagenomics metatranscriptomics of skin swabs and sc-RNAseq of PBMCs. They demonstrate differences in cell types abundance and inflammatory and KRAS signaling activation in response to spaceflight.

The study is underpowered and primarily descriptive, with little biological insights emerging and no experimental validation of the claims in vitro/in vivo/ex vivo. It also appears by the claims of this paper that the authors do not have a deep understanding of fundamental cutaneous biology concepts.

Major concerns:

- An important paper from the space biology field also using multi-omics analyses to characterize spaceflight impact is not referenced at all (PMID: 33242417). How do the results of da Silveira et al compare to the findings of this manuscript?

We appreciate the reviewer's comment and suggestion. We have added the publication in the revised manuscript (page 1, line 64, reference #5). Overall, we found the results reported from the publication are very comparable to the findings of this manuscript. For example, the mitochondrial dysfunction associated with cardiovascular changes discussed in Silveira et al. paper is consistent with our observations in skin vasculature ROIs as discussed in the context of cardiovascular microvascular adaptations and immune changes.

To make this comparison more apparent, in the revised manuscript, we have added comparisons of skin spatial transcriptomics and PBMC single cell data to the twin study blood sequencing data (**Extended Data Fig. 4**). Although the measurements were performed differently, we found consistent changes in pathways we highlighted in the manuscript. Pathways related to inflammation were shown much stronger in our spatial and single cell multiome transcriptomics datasets, and this may be coming from increased sample size and sequencing depth. The updated portion of the manuscript is also copied below (page 5, lines 161-164):

Comparing the pathway-level changes to blood sequencing datasets from the same mission and previous mission (NASA Twin Study, although with different duration of exposure), we found consistent changes in pathways such as KRAS signaling, epithelial-to-mesenchymal transition, and angiogenesis (**Extended Data Fig. 4d**).

- KRAS involvement has previously been shown in spaceflight (PMID: 32045996); also not referenced by the authors.

We thank the reviewer for providing this reference. The publication has been discussed and cited in the discussion section of the revised manuscript (page 8, line 316, reference #51).

- Can the authors explain the time point selection? why 44 days before the mission and then 1 day after?

Time point selection was heavily dependent on the availability (i.e., collection should happen within their schedule but not in conflict with other training sessions) and accessibility (i.e., researchers and clinicians need to meet with crew members to perform biopsies and collect samples) to the crew members. Although the experimental design is very constrained by the crew members' schedules and sample availability, we would like to emphasize this is the first astronaut dataset from biopsied samples and one of the first attempts to standardize sample collection across multiple timepoints and assays.

- Were biopsies taken from sun-protected regions? Same arms? How close were the pre- to post-spaceflight ones. Age/sex of spaceflight crew?

The biopsies were taken from the anatomically same, matched locations. Specifically, the samples are from the lateral upper arm (deltoid) region, so they were relatively protected compared to forearms or wrists, which are more exposed to sunlight. Except for one crew member, the preflight biopsies were taken from the right deltoid region, and the postflight biopsies were taken from the left deltoid region by a dermatologist. For the biopsies that were taken from the same arm, they were approximately 1.5 inches away from one another. We had two females and males each, and the age range was 31-52 (averaged 41.5). We have updated the details of the sample collections in the revised manuscript, and shown below (page 11, lines 433-437):

The biopsies were collected from all four crew members' right (preflight) and left (postflight) deltoid (or upper arm) that were anatomically the same, matched locations, except for one crew member. For this crew member, both timepoints were collected from the right arm, approximately 1.5 inches away from one another. The average age of the crew members was 41.5 and consisted of two males and two females.

- Since freeform ROI selection was completed and it appears from Fig 1a that quite large regions were selected (e.g. for the dermis and epidermis), were all ROIs from the same patient and anatomical location distinct or was there overlap and how did the authors ensure no overlap? Could the authors show for a representative study subject all the ROIs selected?

To avoid overlap of the anatomical location, we avoided the boundary region of each ROI types, for example, we left a gap between epidermis-dermis regions so that we do not accidentally collect signals from the other region. In **Extended Data Figure 1**, we have added all of the ROIs and tissue images from all subjects. In addition, We have updated the detailed explanations of the freeform ROI selections in the methods section (page 13, lines 486-489), also shown below:

OE ROIs covered spinous and granular layers, while IE ROIs covered the basal layer, identified from the staining of the tissue. To ensure proper selection and to avoid overlaps

across different ROI types, small gaps between each ROI type were made, as shown in **Extended Data Fig. 1**.

- how was batch effect corrected?

Although the samples were collected at different timepoints, the tissue processing and GeoMx run was done together to minimize any potential batch effect. We did not see apparent batch effects between the subjects, timepoints, or runs, as shown in the UMAP projection of all ROIs in the analysis (**Fig. 1b**).

- The authors in multiple instances report genes and cell types that are impossible to find in the anatomical locations mentioned. Line 136 “Loss of keratin family transcripts... found in the dermal layer”? could this be explained by a flaw in the selection of ROIs where epidermal regions were included in the pre-flight samples? no keratins are normally expected in the dermis. Line 157 "loss of melanocytes in the OE" Melanocytes are only present in the basal epidermis. Line 160 “fibroblasts in the OE” no fibroblasts are present in the epidermis.

Line 178, reticular fibroblasts are not differentiated fibroblasts

Fig. 3b showing lymphatic EC, pericytes and fibroblasts in the OE, or melanocytes in the VA is incorrect.

Also Extended Fig. 3a shows proportions of cells that are impossible to find in the described compartments. Fibroblasts even in the outer epidermis? Melanocytes throughout the skin?

Line 224 “In the OE... increased lymphatic EC and decrease in endothelial cells”. There are no vessels in the epidermis, especially the outer epidermis.

All these claims seriously compromise the biological validity of the study.

We understand the reviewer’s concern and agree that we observe slight noise in the signals from the data because we are selecting specific tissue layers for characterization. Furthermore, we also agree that there is an inevitable compromise between resolution and strength of the signal, limited by the current technology. To ensure biological validity of the study, we have included similar observations in the literature. We also have been performing comparative studies with other missions and discussed them in the revised manuscript.

We also want to emphasize that these cellular changes are from the cell type deconvolution algorithms that estimates the cell composition of the given ROI. Identification of these signatures in the epidermis does not necessarily mean that fibroblasts, endothelial cells and/or pericytes are actually present in the epidermis. For example, we are aware that there is no fibroblast in the epidermis, but we are seeing fibroblast-associated gene signatures changing in those ROIs. These signature-level changes potentially are related to fibroblast-epithelial cells in epidermis interactions, wound healing response, and extracellular matrix generation that impacts the epidermis as well. Likewise, lymphatic endothelial cell signatures can also be immune-associated signals that reach via microvascular structures that span up to the dermal area.

In addition, although we are aware that many KRT markers are used to identify epidermal regions and have much stronger expression in the epidermis, we found a weak but positive signal of KRT genes in dermis as well. We believe the signal is coming from some of the hair follicles and glands extended into the dermis region. Similarly, because we are getting signals from the hair follicles, we are getting melanocyte-associated signals in dermis region as well.

With this clarifications in mind, in the revised manuscript, we corrected and clarified our statements associated with specific cell types, including fibroblast, lymphatic EC, and reticular fibroblast.

- ATAC-seq is mentioned but not shown?

We appreciate the reviewer's comment. We performed multiome, which allows us to characterize RNA and ATAC profiling on the same cell, on the PBMCs; however, in the original manuscript, we only used RNA signals to compare findings from the skin spatial transcriptomics to the findings from the PBMCs. In the revised manuscript, we show chromatin accessibility changes of the overlapping genes. From chromatin accessibility data, we found that the increased gene activity of the upregulated DEGs found from R+1 timepoints stay longer than RNA expression levels. This result is shown in **Fig. 5d**, and the updated manuscript related to this question is also copied below (page 7, lines 277-280):

While all these overlapping DEGs were temporary in PBMCs, i.e., upregulated in the immediate postflight samples (R+1 timepoint) and returned to pre-flight expression levels, the chromatin accessibility of these genes stayed slightly longer, up to R+45 timepoint (**Fig. 5d**).

- Line 333, what do you mean by "skin cells" ? endothelial cells are also skin cells

We thank the reviewer for pointing this out. We considered endothelial cells as cells around the vasculatures, whereas skin cells include keratinocytes and melanocytes. To clarify, in the revised manuscript, we have corrected this to epithelial vs. endothelial cells (page 10, line 362).

- Line 442 "custom gene signatures obtained.." can the authors provide these signatures, or at least reference the studies that used them ?

We had the reference in the main text where we described the signatures (reference #25-26: Zou *et al.*, 2021 and Solé-Boldo *et al.*, 2020), but in the revised manuscript we added the references (page 13, line 495). All custom gene signatures are shared on github repository as a rds file.

Minor comments:

- Please clarify (L-44) and (R+1) terminology for readers unfamiliar with space travel terms

We have updated our manuscript to clarify the terminology (page 3, lines 88-91), copied below:

We comprehensively profiled skin microenvironment changes in response to spaceflight by performing a multi omics analysis using 4 mm skin punch biopsies from the crew

members (n=4) 44 days before launch (L-44) and one day after return (R+1) of the 3-day mission.

-Line 421, type of slide used? Also how thick were the cryosections?

In the methods section of the revised manuscript contains more detailed information (page 12, lines 469-470) and shown below:

Tissues were then cryosectioned at 5 μm thickness and attached to glass microscope slides (Fisher Scientific, cat# 22-037-246).

Line 252, please correct to “atoPic dermatitis”

Corrected.

Reviewer #2

(Remarks to the Author):

The manuscript by Park et al. titled “Spatial multi-omics of human skin reveals KRAS and inflammatory responses to spaceflight” aims to describe the changes that spaceflight causes to the skin (different sites), using GeoMx™ Digital Spatial Profiler and multi omics assays (single cell RNA-seq, metagenomics, and metatranscriptomics). The authors describe the increase in inflammatory genes and changes in other immune effectors post-spaceflight. In addition, the authors describe the microbial changes in the skin. While there are important implications to the observations the descriptive nature of the results and the lack of validation experiments require additional experimentation and prevent the publication of the manuscript in its current form.

Comments:

The authors should look for changes in antimicrobial resistance in the skin microbiome using the metagenomic and transcriptomic data.

We thank the reviewers for this insightful comment. From our metagenomics and metatranscriptomics data, we mapped of overall associations (to identify a core set of spaceflight-associated proteins across all swabbed body sites) and found 53 proteins transiently increased in-flight, of which 50.9% were found in at least two of the three swabbed broad microbial environments (skin, oral, and nasal). These proteins were mostly enriched in metagenomic data and involved in microbial defense or competition. Other related proteins that showed increase included phage proteins, pathogenicity island proteins, toxin-antitoxin systems, a farnesyl-diphosphate synthase, the crossover junction endodeoxyribonuclease RuvC, and cell surface proteins. However, most of these changes were transient (found stronger signals in-flight), and were discussed in-depth in the separate paper currently in review at *Nature Microbiology* (Braden Tierney *et al.*).

The authors should also present the virome changes due to spaceflight.

We appreciate the reviewer's suggestion. In the revised manuscript, we have mentioned few viral species that change in abundance. Copied below (page 7, lines 245-252):

In addition, from alignment to known viral assemblies we found statistically significant decrease in abundance of reads associated with those from *Uroviricota* (i.e., *Fromavirus*, *Acadianvirus*, *Armstrongvirus*, *Amginevirus*, *Bixzunavirus*) and *Naldaviricetes* (i.e., *Alphabaculovirus*), and increased abundance of reads associated with those from *Negarnaviricota* (i.e., *Almendravirus*, *Orthotospovirus*) and *Cossaviricota* (i.e., *Betapapillomavirus*, *Betapolyomavirus*) (p-values < 0.05). Virome changes are limited by the depth of the sequencing and skin virome knowledge, however we also report relative abundances of both bacterial and viral species (**Table 3**).

Because of the limited scope of the manuscript, we want to emphasize that deeper level of analyses were performed in the other manuscript currently in revision (Tierney et al., *Nature Microbiology*, in revision).

The authors should discuss the limitations of the study small sample size

We agree with the reviewer that this study is limited by small sample size, and we have discussed it towards the end of the discussion section in the original manuscript. We have expanded the limitations and challenges in the revised manuscript (page 11, lines 404-413) and copied below:

Our study is also limited by four crew members, two timepoints, and a short-term mission. To mediate this, the data has been compared with other research and analog models (Henry Cope and Jonas Elsborg et al., *Nature Communications*, in review). In addition, some limitations are due to the unique challenges associated with the astronaut samples, including: their difficult procurement (biopsy), their rarity (there are few missions being conducted), and limited infrastructure for on-site sample collection and processing. Nonetheless, these first structural and molecular views into the epidermal response to spaceflight highlight key KRAS, immune, and related pathways that can be integrated in the future missions to help researchers understand the long-term effects of the spaceflight stressors and cellular changes we identified in this study.

For the pathology shown please add a pathology score, so the reader without training can assess the changes to the tissue.

We appreciate the reviewer's point. As discussed in the original manuscript, from an expert pathologist's evaluation, we found no significant differences in inflammatory or immune cells, and did not find any pathology- and histology- related changes in the skin biopsies we obtained. All of the changes were on the transcript level. In the revised manuscript, we have clarified this point. However, if the reviewer intended a specific type of evaluation, we would be pleased to include a designated score.

In line 140, the authors describe that Ap3b1 transcript was upregulated. The authors should validate this information via qPCR or western blot.

We appreciate the reviewer's comment and we observed downregulation of the AP3B1 transcript in the postflight samples relative to preflight samples. The samples were very limited and majority of the samples were used for spatial transcriptomics and optimizations, and we do not have enough materials for extraction.

However, to address the reviewer's concern, we used remaining FFPE slides to perform RNA scope and to validate expression changes of the given transcript. Consistent to sequencing data, we saw decreased number of AP3B1 transcript in all four crew members and the average change was statistically significant ($p=0.029$). We have included this in the updated manuscript, which can be found in **Extended Data Figs. 4a-c**.

In line 174 the authors state that there was a decrease in protein production, but the data suggest it is gene expression. Please clarify. Also, IHC or IF for protein cell markers would provide additional validation to the data presented and the single cell data.

The reviewer is correct that we meant decrease in genes related to protein production. We have clarified this point. We already had cytokeratin staining done on the tissues for ROI selections and have quantified the average expressions within the epidermis to provide additional validation to the transcriptomics data presented in the manuscript.

189: Did the swab was taken in the exact spot of the biopsies?

The swab was taken in the exact spot of the biopsies right before the biopsy.

The authors constantly state that certain species were changed due to spaceflight, but the data shows only microbiome changes at the genus level. If metagenomic data was collected the authors should show at least show the top species.

We reported at the genus level to ensure we cover broad range of microbe changes in the original manuscript and included the species level observations in **Table 3**, but we agree with the reviewer's comment. In the revised manuscript, we have added the species level description (page 6, lines 218-230), also copied below:

Gross comparison of bacterial species by family showed decreased abundance in *Actinobacteria* (e.g., *Actinomyces sp000220835*) while increased abundance in *Firmicutes/Bacillota* (e.g., *Peptoniphilus C/B*) and *Proteobacteria/Pseudomonadota* (e.g., *Caulobacter vibrioides*, *Sphingomonas carotinifaciens*, *Roseomonas mucosa/nepalensis*) (**Fig. 4c-d, Extended Data Fig. 6b**). When grouped into genus, several species, including *Cutibacterium* (e.g., *Cutibacterium acnes/avidum/modestum/porci*), *Mycobacterium* (e.g., *Mycobacterium paragordoniae*, *Mycobacterium phocaicum*), and *Pseudomonas* (e.g., *Pseudomonas auruginosa/nitroreducens*) showed statistically significant decrease (p -values < 0.05). Several species including *Streptococcus* (e.g., *Staphylococcus capitis*,

Streptococcus mitis BB) and *Veillonella* (e.g., *Veillonella atypica/parvula/rogosae*) showed significant increase (Fig. 4d). Also, species under the *Staphylococcus* genus, such as *staphylococcus capitis/epidermidis/saprophyticus* showed slight decrease while the relative abundances were highly variable across biological replicates.

Metagenomics could be used to assemble DNA sequences for the microbiome to use the transcriptomic data to be mapped to it. Thus a multiomics analysis of the skin microbiome.

We appreciate the reviewer for this comment. We have done similar analyses in the other manuscript currently in revision (Tierney et al., *Nature Microbiology*, in revision), where we provide rationale for each analysis method, in-depth analysis and different assembly methods we have used to understand microbial changes in both metagenomics and metatranscriptomics data. We believe the scope of this manuscript should be around skin transcriptome changes and factors around it such as microbiome and immune measurements and did not include as detailed information as the manuscript dedicated to metagenomics dataset, however we would be happy to address if there is any specific question/perspective the reviewer think is important.

The paragraph of lines 220 to 233 should be moved to the discussion. It lacks enough data for the stated conclusions.

We agree with the reviewer's suggestion and have moved the paragraph to the discussion (pages 10-11, lines 395-401).

Cytokine data has no protein-based or qPCR validation or validation in the tissue acquired.

As an additional confirmation, we have added cytokine analysis directly from the crew members' serum samples (**Fig 5b**). All of these cytokines are increased in postflight timepoints. Especially IL5 shows the biggest and consistent change in the R+1 time point from skin transcriptomics, cytokine assay, and multiome gene expression data.

The authors should integrate the microbiome data with the immune data (figure 5) and assess what microbial changes correlate with inflammation and immune cell changes.

We thank the reviewer for the suggestion. We have compared the microbiome and immune changes in the other manuscript currently in revision (Tierney et al., *Nature Microbiology*, in revision), and found microbes that are positively and negatively associated with immune cell type specific changes in post-flight samples relative to pre-flight samples. In the revised manuscript, we discussed several major findings and observations from the comparison between immune populations and microbiome; specifically, we have identified associated microbe-gene expression pairs for each spatial layer. The updated manuscript is copied below (page 7, lines 252-256):

To explore microbiota-skin interactions, we also identified potential associations between microbiome shifts from metagenomics/metatranscriptomics data and human gene expression from skin spatial transcriptomics data; these included associations were with viral phyla (i.e., *Uroviricota*, *Cressdnaviricota*, *Phixviricota*), which is a potentially

interesting area to explore as more crew samples are collected. (**Extended Data Fig. 6d-e, Table 3**).

However, we also would like to emphasize that the scope of this manuscript covers around skin regions, and we have an entire manuscript focusing on immune associated changes and its interactions with skin and metagenomics assays.

In extended data figure 2, there seem to be only one sample for C002 for OE and IE. For C002, we only had one ROI each for OE and IE because of the tissue structure we identified from the staining. We have collected ROIs that captures both OE and IE (and labeled OE+IE). We have included this to overall comparisons and epidermal (OE+IE) differential analysis, but not in this particular analysis.

REVIEWERS' COMMENTS

Reviewer #1 (Remarks to the Author):

The changes and additional analyses are appreciated and have improved the manuscript. Especially the confirmatory new cytokine data and comparisons with other relevant studies 'datasets.

Some minor issues to be fixed before publication:

-lines 172-173: replace "fibroblast population is rarely found in OE" with "fibroblast is an unanticipated cell type in the epidermis"

-lines 399-400: the word "changes" is repeated

-line 468: replace "patient" with "subject"

-line 590: specify AP3B1 gene region (currently written as xxx-xxx)

-line 611: make sure <https://github.com/eliah612o/inspiration4-omics> works

Reviewer #2 (Remarks to the Author):

The authors have addressed this reviewer's comments

Response to Reviewers

Reviewer #1 (Remarks to the Author):

The changes and additional analyses are appreciated and have improved the manuscript. Especially the confirmatory new cytokine data and comparisons with other relevant studies 'datasets.

Some minor issues to be fixed before publication:

-lines 172-173: replace “fibroblast population is rarely found in OE” with “fibroblast is an unanticipated cell type in the epidermis”

-lines 399-400: the word “changes” is repeated

-line 468: replace “patient” with “subject”

-line 590: specify AP3B1 gene region (currently written as xxx-xxx)

-line 611: make sure <https://github.com/eliah612o/inspiration4-omics> works

We would like to thank the reviewers for their suggestions, and we have updated our manuscript so that the minor issues that the reviewer pointed out are corrected.

Reviewer #2 (Remarks to the Author):

The authors have addressed this reviewer's comments.

We appreciate the reviewer's comment.